# Emerging Nanotechnology in Non-Surgical Periodontal Therapy in Animal Models: A Systematic Review

**DOI:** 10.3390/nano10071414

**Published:** 2020-07-20

**Authors:** Adrian Brun, Nicolas Moignot, Marie-Laure Colombier, Elisabeth Dursun

**Affiliations:** 1Faculty of Dental Surgery, Université de Paris, CEDEX F-92120 Montrouge, France; adrian.brun@parisdescartes.fr (A.B.); nikaulhas@free.fr (N.M.); marie-laure.colombier@parisdescartes.fr (M.-L.C.); 2Orofacial Pathologies, Imaging and Biotherapies laboratory, UR2496, Université de Paris, F-92120 Montrouge, France; 3Division of Periodontology, Department of Oral Medicine, Henri Mondor Hospital, APHP, F-94000 Créteil, France; 4Department of Oral Medicine, Bretonneau Hospital, APHP, F-75018 Paris, France; 5Division of Periodontology, Department of Oral Medicine, Louis Mourier Hospital, APHP, F-92700 Colombes, France; 6Innovative Dental Materials and Interfaces Research Unit (URB2i), EA4462, Université de Paris, Université Sorbonne Paris Nord, F-92120 Montrouge, France; 7Division of Paediatric Dentistry, Department of Oral Medicine, Henri Mondor Hospital, APHP, F-94000 Créteil, France

**Keywords:** periodontitis, non-surgical periodontal therapy, nanoparticles, nanotechnologies

## Abstract

Periodontitis is one of the most prevalent inflammatory diseases. Its treatment, mostly mechanical and non-surgical, shows limitations. The aim of this systematic review was to investigate the effect of nanoparticles as a treatment alone in non-surgical periodontal therapy in animal models. A systematic search was conducted in Medline/PubMed, Web of Science, The Cochrane Library and Science Direct. The eligibility criteria were: studies (i) using nanoparticles as chemotherapeutic agent or as delivery system; (ii) including preclinical controlled animal model (experimental periodontitis); (iii) reporting alveolar bone loss; (iv) written in English; and (v) published up to June 2019. Risk of bias was evaluated according to the SYstematic Review Centre for Laboratory Animal Experimentation. On the 1324 eligible studies, 11 were included. All reported advantages in using nanoparticles for the treatment of periodontitis, highlighted by a reduction in bone loss. Agents modulating inflammation seem to be more relevant than antibiotics, in terms of efficiency and risk of antibiotic resistance. In addition, poly(lactic-co-glycolic acid) or drugs used as their own carrier appear to be the most interesting nanoparticles in terms of biocompatibility. Risk of bias assessment highlighted many criteria scored as unclear. There are encouraging preclinical data of using nanoparticles as a contribution to the treatment of periodontitis.

## 1. Introduction

Periodontitis is the 6th most prevalent disease worldwide, with an overall prevalence of 11.2% and around 743 million people affected [1]. It is defined as an inflammatory disease of the tooth-supporting tissues, involving polymicrobial synergy and dysbiosis [2]. The inflammation leads to progressive and irreversible destruction of all periodontal tissues, with clinical attachment loss, periodontal pocket formation, bleeding on probing and alveolar bone loss, which eventually results in tooth loss [3].

The gold standard primary treatment for periodontitis is etiological and consists in the mechanical disruption and removal of pathogenic biofilm and calculus from the teeth and surrounding structure, by ultrasonic scaling and root planning, in conjunction with a patient’s proper plaque control [4]. However, some sites and patients may not respond adequately, due to the impossibility to access areas such as furcation or root depression, and the limited effect over specific pathogens invading soft tissues [5]. Adjunctive use of chemotherapeutic agents, such as systemic or local antibiotics or topical antiseptics [6] has been indicated to improve the clinical outcomes in certain patients and periodontal conditions [7], to avoid the need of surgical therapy [6]. The benefits of systemic antimicrobials have been reported in several systematic reviews and meta-analysis [8,9], nevertheless it can lead to side effects and development of resistance or overgrowth of other pathogens [10]. Local application of antibacterial agents into periodontal pockets involves drug delivery systems (irrigating systems, gels, fibers, strips/films, inserts, microparticles and so forth, which would ideally allow an easy reach of the targeted site, a retentivity at an effective concentration and a controlled release for an adequate period of time [11]. However, some issues still exist such as: difficult placement, uncontrolled drug release, rapid local drug clearance and need to remove non-biodegradable forms [12,13].

In the recent years, there has been growing interest in nanotechnologies in medicine and dentistry. Although the ISO norm defines nanoparticles as particles between 1 and 100 nm, there are discussions about its definition, that may vary according to the discipline. In drug delivery, i.e., pharmaceutical area, nanomaterials are defined as having a size between 1 and 1000 nm (FDA Guidance 2014) [14]. Engineered particles may be used as carrier, but also a drug itself may be formulated at a nanoscale and act as its own carrier [15]. They may be of biological origin like phospholipids, lipids, lactic acid, dextran, chitosan or of chemical origin like various polymers, carbon, silica, and metals.

Owing to their small size and high surface area, nanosized particles show higher chemical reactivity and increased electrical, magnetic, optical, biological or mechanical properties, than their macroscopic or microscopic counterparts [16]. In particular, intensive research focuses on the development of advanced drug delivery systems. Nanosized particles can contain a wide range of substances, while accurately delivering the active therapeutic agents to the target site and easily penetrating regions inaccessible to other delivery systems, such as deep periodontal pockets [17]. Their increased stability and controlled release ability allow maintaining local effective drug concentration for a longer time, which may reduce the frequency of administration and the drug dose [18,19].

Various nanoparticulate systems have been investigated for the treatment of periodontitis [20,21]. Despite the large number of studies, there is no comprehensive and updated systematic review reporting the available evidence and/or indications for future research. Moreover, animal studies are considered as crucial for further human clinical trials. Hence, the aim of this study was to systematically review the effect of nanotechnology-based drug systems as a treatment alone in the non-surgical treatment of periodontitis in preclinical animal models. Agents modulating inflammation seem to be more relevant than antibiotics, in terms of efficiency and risk of antibiotic resistance. In addition, poly(lactic-co-glycolic acid) or drugs used as their own carrier appear to be the most interesting nanoparticles in terms of biocompatibility.

## 2. Materials and Methods

### 2.1. Protocol

This systematic review was conducted in accordance with the Preferred Reporting Items for Systematic Review and Meta-Analysis (PRISMA) guidelines [22]. The protocol was registered in PROSPERO (International Prospective Register of Systematic Reviews), at the UK’s National Institute for Health Research (NHS), University of York, Centre for Reviews and Dissemination, under the number: CRD42019134805. To structure the research question “Are nanoparticles useful in non-surgical periodontal treatment?” the PICO model was applied [23].
Participants/population: animals with experimental periodontitisIntervention/exposure: use of nanoparticles systems as non-surgical periodontal therapy (i.e., injected in the periodontal pocket without involving incisions and flap elevation)Comparator/control: untreated animalsOutcome: decrease of alveolar bone loss

### 2.2. Search Strategy for the Identification of the Studies

A comprehensive literature search was performed on the electronic databases Medline via PubMed, Web of Science, the Cochrane Library and ScienceDirect, up to June 2019. The search equations were built by combining keywords related to periodontitis with keywords related to nanotechnology (Appendix A). In addition, a manual screening was carried out among the references of selected articles in order to gather further relevant papers.

Screening of titles and abstracts was conducted by two independent reviewers (A.B., N.M.). Irrelevant studies, i.e., unrelated to the treatment of periodontitis with nanotechnology, were excluded. Then, full texts were assessed according to inclusion criteria. In case of disagreement, a consensus was obtained with a supervisor (E.D.). Reasons for study exclusion were also recorded.

### 2.3. Eligibility Criteria

The following inclusion criteria were applied during the literature search: (i) studies using nanoparticles as chemotherapeutic agent or as delivery system; (ii) preclinical controlled animal model studies: all animal species with experimental periodontitis, treated by non-surgical periodontal therapy; (iii) studies reporting alveolar bone loss; (iv) studies written in English language; and (v) studies published up to June 2019.

The following exclusion criteria were: (i) irrelevant studies (not dealing with nanoparticles nor periodontitis); (ii) studies involving surgical periodontal treatment; (iii) studies without control group (i.e., studies testing only one group using nanoparticles in only one way) or with methodological issues; (iv) if it was not previously evaluated in the literature, studies without assessment of nanoparticles cytotoxicity; and (v) case reports, case series, reviews, in vitro and ex vivo studies.

### 2.4. Outcome Measures

The primary outcome measure was a decrease of alveolar bone loss (ABL) evaluated macroscopically, radiographically (µCT (computed tomography)) or microscopically (dissecting microscope). The secondary outcomes were an improvement of bactericidal effect and/or a decrease in inflammatory response, including the measure of anti-inflammatory cells or mediators’ level.

### 2.5. Data Extraction and Analysis

For each paper, the following data were extracted independently and in duplicate by the two reviewers (A.B., N.M.): treatment type, authors and year, drug tested, study design and models, measures, objectives and main results. Data were collected in an excel spread sheet. In case of disagreement regarding the extraction of data, a consensus was obtained with the supervisor (E.D.).

### 2.6. Study Quality Assessment

The risk of bias was independently evaluated by the two reviewers (A.B., N.M.) according to the SYstematic Review Centre for Laboratory Animal Experimentation (SYRCLE) [24] risk of bias tool for the experiments reported in the studies. Disagreements about rating were resolved by a third reviewer (E.D.). SYRCLE is derived from the Cochrane’s Risk of Bias tool for clinical studies [25], and it was adapted to be applied to animal studies. The tool consists of 10 main questions related to selection bias, performance bias, detection bias, attrition bias, reporting bias and other biases. The responses to the tool’s questions were answered as “Yes” (question adequately answered), “No” (question not answered) or “Unclear” (not enough information to answer yes or no). Based on the answers to the signaling questions, the risk of bias domains were classified as low, high or unclear. An overall risk was evaluated.

## 3. Results

### 3.1. Study Selection

A total of 1324 eligible studies were identified in the electronic databases: 558 from Medline via Pubmed, 516 from Science direct, 249 from the Web of Science and 1 from the Cochrane Library. Manual search did not determine any further study for inclusion. The last search date was in June 2019. After exclusion due to duplication and titles and abstracts review, 21 articles were selected. After full reading, 10 articles were excluded (Appendix A). Finally, 11 studies were included: 2 with antibacterial agents [26,27], 8 with anti-inflammatory agents [28,29,30,31,32,33,34,35] and 1 with both [36]. The process of article selection is summarized in Figure 1.

### 3.2. Characteristics of the Included Studies

All relevant information related to study characteristics and data synthesis are presented in Table 1. All included studies were in vivo animal studies, published between 2010 and 2019. The follow-up of the studies was up to 28 days. Two studies used mice [27,28] and 9 used rats [26,29,30,31,32,33,34,35,36] (including one with systemically compromised animals with diabetes mellitus [32]). Number of animals ranged between 4 and 20 per group. In five studies, ligatures were used to induce periodontitis [26,29,32,34,36], in six studies, disease was induced by bacteria or lipopolysaccharides (LPS) [27,28,30,31,33,35]. The involved teeth were either the first and/or the second molar in the upper and/or lower jaw.

### 3.3. Nanoparticles Type

All relevant information related to nanoparticles characteristics are presented in Table 2. Among the 11 included studies, 7 different nanoparticles were tested: nanostructured doxycycline gel [26], poly(ethylene glycol)-poly(lactic acid) copolymer (PEG-PLA) [35], poly(lactic-co-glycolic acid) (PLGA) [27,28,32,33,36], nanoemulgel with eugenol as oil phase [29], polyon complex composed of triblock copolymer (PMNT-PEG-PMNT) and anionic poly(acrylic acid) [30], zinc-hydroxyapatite (chitosan-based) [31] and gold (Au) [34]. In two studies nanoparticles were associated with an antibacterial agent (BAR peptide [27]) or developed from an antibacterial agent (doxycycline [26]). Eight other studies were interested in the anti-inflammatory effect of the nanoparticle [30,31] or of the anti-inflammatory agent coated on/loaded in the nanoparticles (15-Deoxy-D12,14-PG J2 [28], ketoprofen [29], metformine hydrochloride [32], curcumin [33], L-cysteine [34], and auranofin [35]). The last study used both antibacterial (metronidazole) and anti-inflammatory agents (*N*-phenacylthiazolium bromide) [36]. Most of the nanoparticles were spherical [26,27,28,29,32,35,36], whereas only one was flower-like micelle [30]. Three studies did not give information about nanoparticle shape [31,33,34]. The final product size was between 45 and 499 nm, but it was not addressed in four studies. [28,31,33,35] The dose of the product was only specified in five studies [26,27,31,34,35]. Four studies reported good cytocompatibility of the tested product [27,33,34,35], four others referred to previous published studies for cytotoxicity evaluation [29,30,32,36], and three did not address the issue [26,28,31] but used nanoparticles that have already been reported to have good cytocompatibility. Most of the tested nanoparticles were biodegradable [27,28,32,33,35,36], whereas only one nanoparticle was non-degradable [34]. Nevertheless, the degradability of the product was not addressed in five studies, but they used nanoparticles that have already been reported to be biodegradable.

### 3.4. Reported Outcomes

Only one murine model study measured clinical parameters [29]. In the 11 murine model studies, alveolar bone level/amount was evaluated macroscopically [26], radiographically or microscopically: 7 studies used µCT (computed tomography) analysis [30,31,32,33,34,35,36] and 3 studies used a dissecting microscope [27,28,29]. Only one study evaluated bone mineral density [31] and two other ones evaluated alveolar bone roughness [26,29]. Fluorescence in vivo imaging and gingival blood flow measures were performed to follow the nanoparticles in the gingival sulcus [30]. Besides, Elisa immunoassays were implemented to evaluate inflammatory response (IL (interleukin)-1β [29,30,32], TNF (tumor necrosis factor)-α [29,32], malondialdehyde [30], IL-6 [30], MPO (myeloperoxidase) [28]), bone resorption (bone alkaline phosphatase, OCN (osteocalcin), tartrate-resistant acid phosphatase 5b) [31] and specific immune response (Ig (immunoglobulin) G anti-*Actinobacillus actinomycetemcomitans*) [28]. Western blots were also performed to evaluate inflammatory response (MAPK (mitogen-activated protein kinase) [33], NF-KB (nuclear factor kappa B) [33], CD (cluster of differentiation) 55 [28]) and bone resorption (RANKL (receptor activator of nuclear factor kappa-B ligand), OPG (osteroprotegerin)) [28]. qRT-PCR (AMP-activated protein kinase, NF-KB p65, high-mobility group box 1, and TGF-b–activated kinase 1 [32]; or IL-6, IL-10, IL-15, IL-17, CCL (C-C motif chemokine ligand)-22, FOX (forkhead box) P3 and TGF (transforming growth factor)-β [28]) was implemented to evaluate inflammatory response. Flow cytometry was used to evaluate lymphocytes infiltration in submandibular lymph nodes (CD4, CD8, CD25, FOXP3) [28]. Finally, histological assessment was used to analyze the architecture of the tissue [26,27,28,29,30,31,32,33,34,36], to count inflammatory cells [29,32,33,34,36], to evaluate inflammatory response (IL-17 [27], inducible nitric oxide synthase [34]), or to count bone cells [30,32,33,34] (tartrate-resistant acid phosphatase, RANKL, cathepsin K, OPG, OCN).

### 3.5. Main Results

All the included studies reported advantages in using nanoparticles for the treatment of periodontitis. One study reported lower GI and TM in rats [29]. All the studies reported a decrease in ABL (expressed in mm or in %) and 3 studies [30,33,34] also showed a decrease in osteoclast count. Two studies showed a reduction of roughness [26,29] and one study, a higher bone mineral density [31]. However, one study reported no change in OPG level [28]. Seven studies showed less inflammatory cell infiltration or lower inflammatory mediators [27,28,29,32,33,34,36] and one study showed lower submandibular nodes infiltration by higher lymphocytes and CD55 expression [28]. However, one study reported no changes in the levels of IL-1β and IL-6 despite a significant suppression of oxidative stress [30]. One study showed no change in IgG anti-*Actinobacillus actinomycetemcomitans* level [28]. None of the studies actually compare the use of nanoparticles to the gold standard periodontal treatment. No studies compared the efficacy of different nanoparticles.

### 3.6. Risk of Bias Assessment

Risk of bias assessment using SYRCLE’s tool led to a total of 110 entries (Figure 2). Two studies were assessed as high risk of bias, because of incomplete outcome data and/or selective outcome reporting [26,30]. The nine other studies [27,28,29,31,32,33,34,35,36] were assessed as moderate risk of bias because no item was judged as high risk of bias whereas most of the items were judged as moderate risk of bias. More specifically, among the 110 entries, 38.2% were answered as “Yes”, 2.7% as “No” and the remaining 59.1% as “Unclear”; this means that about most of the criteria which are considered as relevant for the reporting of preclinical trials were not reported. Reporting of baseline characteristics, selective outcome reporting and consistency of statistical analysis were frequently judged as of low risk of bias (81.8–100.0%); sequence generation, allocation concealment, random housing and blinding of the investigators and caregivers, random outcome assessment and blinding and incomplete outcome data were often judged as of unclear risk of bias (63.6–100.0%). None of the items were mostly judged as of high risk of bias (0.0–18.2%). Risk of bias in the included studies is summarized in Figure 3.

## 4. Discussion

### 4.1. Application of Nanotechnology in Periodontology

The included studies highlighted the growing interest in nanotechnology for periodontal treatment. In fact, the first study was published in 2010, six were published between 2010 and 2017, and five were published between 2018 and June 2019. Engineered particles may be used as carrier, but also a drug itself may be formulated at a nanoscale and act as its own carrier.

The data of studies were compared as possible. ABL was analyzed when it was expressed in mm. The overall results of the present systematic review showed that nanoparticles may have a positive effect to prevent alveolar bone loss in periodontitis animals. Various types were used in the included studies, each with advantages and relative drawbacks. For example, PLGA has a modulable viscosity but has an initial burst release. Chitosan is biologically active and has good antibacterial properties but is also toxic at high concentration. However, all have the following important characteristics: biodegradability, biocompatibility, autoregulation of drug release rate and inherently immunogenic by the ability to display multiple antigens [37].

Besides, none of the studies compared the use of nanoparticles to the gold standard of periodontal treatment, i.e., mechanical debridement, alone or associated with chemotherapeutic agents, which would attest the benefit of nanoparticles. Moreover, the lack of reporting on methodology led to “Unclear” risk of bias for most of the main questions of SYRCLE’s tool. Finally, very little was discussed about the safety of using nanoparticles.

### 4.2. Nanotechnologies and Antibacterial Agents

Periodontitis is induced by certain bacteria. This is the reason why antibacterial agents may promote periodontal healing. Among the 11 included studies, 3 used antibacterial agents [26,27,36]. Only one study evaluated in vitro the antibacterial effect of the agent [27] and was considered as efficient. However, the complexity of the 3D organization of the subgingival biofilm cannot be faithfully reproduced in vitro because of the heterogeneity of its composition, Moreover, antibacterial activity was not tested in the two other studies, which makes impossible to highlight the superiority of one antibacterial agent to another.

All the studies reported a decrease of periodontal destruction in measuring ABL. One study [26] used doxycycline and also reported a significant decrease in inflammatory cell activation (by measuring MPO), as well as lower roughness. Another one [36] used metronidazole and also showed reduced inflammation compared to non-treated periodontitis after 21 days. However, even if bone loss was significantly reduced after 4 days, it was not after 21 days. Finally, a last one [27] investigated the role of a BAR-peptide, which is derived from bacteria and inhibits adherence between *P. gingivalis (Porphyromonas gingivalis)* and *S. gordonii* (*Streptococcus gordonii)*, and also *P. gingivalis* virulence. It also reported lower IL-17 expression. IL-17 is precisely a key cytokine that allow neutrophil recruitment and has also a potent pro-osteoclastogenic effect contributing likely to the pathogenesis of periodontitis [38].

Even if the combination of metronidazole and amoxicillin in systemic uptake is the most effective antibacterial adjuvant in treating periodontitis [7], local application of tetracyclines was also used for its anti-inflammatory properties at subantimicrobial dose and showed interesting results. The combination of metronidazole and amoxicillin was not tested in the included studies, whereas one study involved tetracyclines, at a subantimicrobial dose [39].

It was difficult to compare the ABL results between the studies, because of many various parameters, however, we can note that the difference in ABL between treatment and control group, is higher for BAR peptide and doxycycline than metronidazole. One could argue that specific-targeted antimicrobial or antimicrobial also having anti-inflammatory properties should be preferred. Besides, despite some moderate clinical benefit and improvement, the interest of using antibiotics is decreased by the risk of antibiotic resistance.

### 4.3. Nanotechnologies and Host Modulation Therapies

Periodontitis is today considered as a dysbiotic inflammatory disease, underlying the breakdown of host-microbe periodontal homeostasis [2]. Hence, host modulation therapies would be relevant and have also been studied as improving periodontal treatment [40]. However, nonsteroidal anti-inflammatory drugs cause unwanted effects, including gastrointestinal, renal or haemostatic effects, as well as hypersensitivity reactions; and steroids are also associated with a range of side effects (including indigestion, altered carbohydrate and protein metabolism, osteoporosis or immunosuppression).

Among the 11 included studies, 9 studies used anti-inflammatory agents, 7 coated on/loaded in the nanoparticles and 2 with anti-inflammatory agents as nanoparticles [30,31]. All these studies showed a clinically significant reduction in ABL. Three studies showed lower osteoclast count [30,33,34]. Inflammatory response was evaluated by the analysis of a lot of cells and mediators. However, the inflammatory mediators the most involved in periodontitis [41,42], IL-1β, IL-6, and TNFα were measured, even partially measured, only in 4 studies [28,29,30,32]. All the studies showed a decrease of inflammatory response, but interestingly, in one of these studies [30], no significant changes were observed in the levels of IL-1β and IL-6. The studies investigating other mediators, such as MPO, MAPK, NF-Kb, CCL-22, FOXP3, did not justify their choice. Moreover, as anti-inflammatory activity was not similarly tested in all the studies, it is impossible to highlight the superiority of one anti-inflammatory agent to another.

It was difficult to compare the ABL results between the studies investigating antimicrobial and anti-inflammatory agents, because of many various parameters, however, ABL seems to be generally lower when using anti-inflammatory. Moreover, one study [36] underlined the significant best results obtained with a host-modulator, compared with an antibacterial, at 21 days compared to 4 days. Agents promoting matrix deposition may also help the longer-term repair process.

A lot of anti-inflammatory agents are available, steroidal and non-steroidal anti-inflammatory agents, as well as other molecules, such as anti-cytokines, histone deacetylase inhibitors or pro-resolving lipid mediators [43]. It would be relevant to test and compare their efficiency in periodontal pockets.

### 4.4. Toxicity

Among the 11 included studies, only 3 studies clearly reported toxicologic data [27,34,35] and, each of them, according to various protocols. Two of them used more than one assay [27,34]. Toxicity was evaluated between 24 and 48 h. The other studies referred to the literature or did not address this subject. Ideally, cytotoxicity assays should be conducted for a period equal to the degradation time of nanoparticles or at least up to the duration of cells survival. However, none of the included studies implemented a long-term cytotoxicity evaluation. Moreover, no studies reported 100% of cell viability. When applying nanoparticles in the periodontal pocket and over time, nanoparticles may be swallowed by the patient. Thus, cytotoxicity assays should involve other cell types, such as the border cells of the digestive tract, but none of the included studies mentioned this subject.

Only one nanoparticle was not biodegradable [34]. In five studies, biodegradability was not discussed. Furthermore, none of the studies mentioned a possible effect of the association of a drug on the biodegradability of the nanoparticles.

According to the rise of nanotechnology in the medical field, the evaluation of the toxicity of nanomaterials is crucial. Indeed, previous research reported potential effects due to their dissemination in the body [44,45]. Besides, for the same target tissue, the toxicity of a nanomaterial is likely to vary according to physicochemical properties, galenical used, dose administered and exposure time. To conclude on the safety of nanomaterials for periodontal use, further toxicological studies must be carried out, according to a standardized protocol, ideally including more than one assay for a better reliability [46,47].

### 4.5. Methodological Heterogeneity and Limitations

A meta-analysis was not possible due to the difference of results measurement and reporting. Indeed, the evaluation of the ABL was different (either macroscopically, radiographically with µCT (computed tomography) analysis or microscopically with a dissecting microscope). Some studies used exact numerical values, whereas some used percentages and others only present diagrams with imprecise scale. Indeed, we had to manually measure the data on the diagrams.

In fact, they presented many differences regarding the protocol of experimental periodontitis, the nanoparticles chosen, the therapeutic agents, as well as the studied parameters.

Regarding experimental periodontitis, mice and rats were used. However, each species has its anatomical and biological characteristics that lead to differences in the periodontitis. Furthermore, five studies used ligatures to induce periodontitis and six studies used bacteria or LPS. A difference in the timing of the evolution of the periodontitis has been described between ligatures and bacterial use [48]. Among the six studies using bacteria, three studies injected them into the gingival connective tissue, two studies placed them in oral cavity, and for the last one it was unspecified. Thus, the induced periodontitis was different: in case of injection, the involvement of sulcular and junctional epithelium was bypassed. Moreover, various bacteria were chosen (especially *Actinobacillus actinomycetemcomitans* [28,35], *P. gingivalis* [27,30,31], *Escherichia Coli* [33]) sometimes associated with other bacteria.

Regarding the nanoparticles chosen, seven different particles have been investigated. It would be relevant to test different nanoparticles in the same study in order to compare their efficacy. Besides, lack of information about nanoparticles form and inoculation also makes comparisons impossible.

Regarding the studied parameters, as widely detailed above, too many variables were studied, hindering comparisons.

### 4.6. Risk of Bias

The risk of bias assessment was mostly “Unclear”, according to SYRCLE’s guidelines, because of lack of reported methodological information, especially regarding randomization. More specifically, no studies explicitly mentioned a sequence generation, and only 36.4% of the studies [26,31,34,36] mentioned allocation concealment, but it can be assumed that these items were probably respected. Besides, the respect of strict animal housing conditions is important to avoid major risk of performance bias. For example, increased gingival inflammation and ABL have already been described as a consequence of sleep deprivation or stress [49]. In the selected studies, baseline characteristics were often described (90.9%), but only 9.1% of the studies mentioned random housing [33] and no studies reported blind housing. Randomization of the samples during the analysis of the results is also crucial to avoid major detection bias. However, the random animal selection for outcome assessment [26,36] and blinded outcome [26,32,33] assessors were poorly described (18.2–27.3%). Finally, statistical analysis was well detailed in the selected studies. Moreover, two studies did not discuss all the outcome data [26,30]. Thus, SYRCLE guidelines should be followed and special consideration should be given to randomization protocols, technician manipulation and animal housing facilities, which could favor homogenization in animal model trials allowing for proper assessment and synthesis of the results.

## 5. Conclusions

The results herein show that nanoparticles used in periodontal indications may have a positive effect on alveolar bone loss in preclinical studies. High nanoparticles sustainability with an extended release would be of crucial interest. The agents modulating inflammation seem to be more relevant than antibiotics, in terms of efficiency and risk of antibiotic resistance. Moreover, poly(lactic-co-glycolic acid) or drugs used as their own carrier appear to be the most interesting nanoparticles in terms of biocompatibility.

Finally, despite lack of strong evidence on their clinical efficacy, there are encouraging preclinical data of using nanoparticles as a contribution to the treatment of periodontitis. Further evidence is however needed due to the safety concerns.

## Figures and Tables

**Figure 1 nanomaterials-10-01414-f001:**
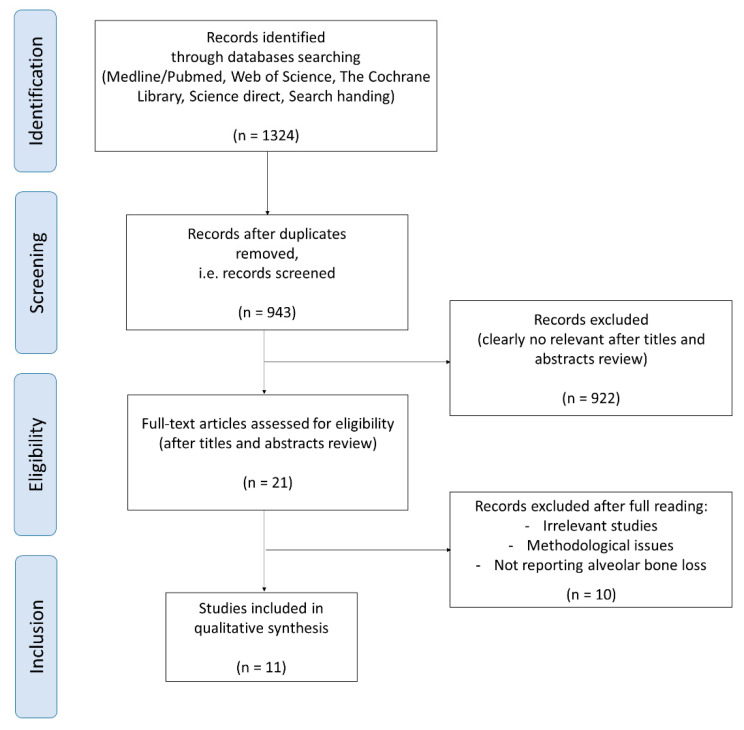
Flow diagram of the manuscript selection process.

**Figure 2 nanomaterials-10-01414-f002:**
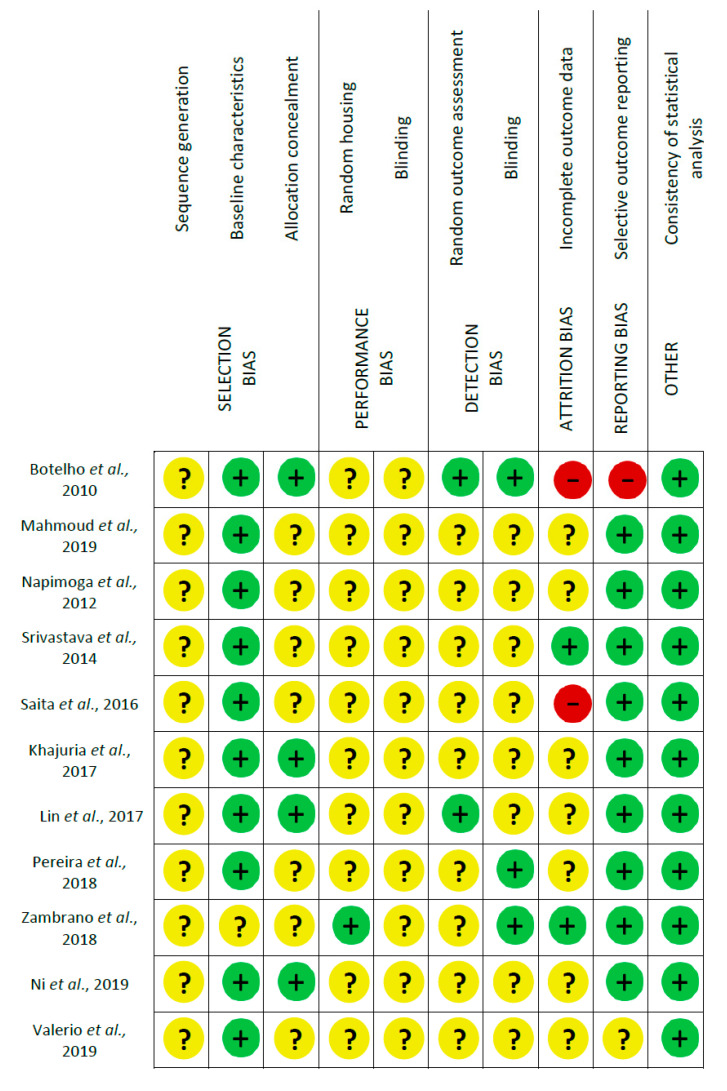
Risk of bias assessment evaluated according to the SYstematic Review Centre for Laboratory Animal Experimentation (SYRCLE): authors’ judgment about each risk of bias item (green = low, yellow = moderate, red = high).

**Figure 3 nanomaterials-10-01414-f003:**
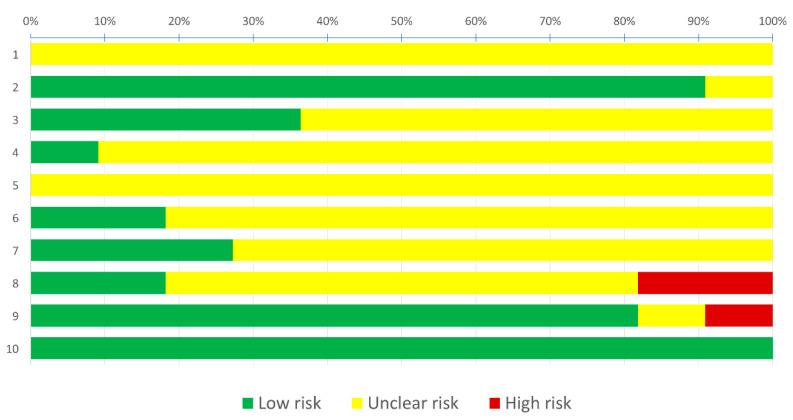
Risk of bias assessment evaluated according to the SYstematic Review Centre for Laboratory Animal Experimentation (SYRCLE): authors’ judgment about each risk of bias item presented as proportions. Selection bias: (1) sequence generation; (2) baseline characteristics; (3) allocation concealment. Performance bias: (4) random housing; (5) blinding. Detection bias: (6) random outcome assessment; (7) blinding. Attrition bias: (8) incomplete outcome data. Reporting bias: (9) selective outcome reporting. Other: (10) consistency of statistical analysis.

**Table 1 nanomaterials-10-01414-t001:** The included studies: main characteristics, objectives and results.

Treatment Type	Authors, Year	Drug Tested	Study Design/Models	Measures	Objectives	Main Results
Antibacterial agent	Botelho et al., 2010 [26]	Nanostructured doxycycline gel	Experimental periodontitis (EPD) in rats induced by ligatures. 4 groups (*n* = 6/group): naïve (no EPD), non-treated (NT), vehicle gel (VG), and DOX	Macroscopic evaluation	Test the efficacy of a locally applied 8.5% nanostructured doxycycline (DOX) gel in preventing alveolar bone loss (ABL)	After 11 days DOX gel reduced the ABL (0.1 mm), compared to the NT (0.7 mm) (*p* < 0.05) and the VG groups (0.4 mm) After 6 h, MPO activity in DOX group (8.5 U/mg) was reduced (*p* < 0.05) compared to NT group (36 U/mg) AFM showed several grooves were observed on the surface of the alveolar bone and other periodontal structures in the NT and VG groups, with significantly greater depths compared to the DOX group (*p* < 0.05)
				Histopathological analysis		
				Myeloperoxydase (MPO)		
				Atomic force microscopy (AFM)		
	Mahmoud et al., 2019 [27]	BAR-modified nanoparticles (BNP) of poly(lactic-co-glycolic acid) (PLGA)	Experimental periodontitis in mice induced by oral delivering of *S. gordonii* and *P. gingivalis* 7 groups (*n* = 8/group): control (without infection), *S. gordonii* only, *P. gingivalis* only, 0.7 µM BAR, 3.4 µM BAR, 0.7 BNP	Dissecting microscope fitted with a video imaging marker measurement system	Determine the efficacy of BNP by inhibiting the adhesion of *P. gingivalis* to *S. gordonii*	Treatment of *P. gingivalis/S. gordonii* infected mice with BNPs reduced bone loss (−0.24 ± 0.05 mm) compare to sham-infected mice (−1.37 ± 0.31 mm, *p* ≤ 0.0001) and to a greater extent than treatment with 0.7 or 3.4 μM free BAR (−0.69 ± 0.1mm, −0.56 ± 0.09mm, *p* ≤ 0.0001) BNP also reduced IL-17 expression almost to the levels of sham-infected mice, whereas the gingival tissue of *P. gingivalis/S. gordonii* infected mice and mice treated with 0.7 µM free BAR demonstrated a statistically significant increase in IL-17 gingival tissue fluorescence (≈2 fold more) compared to uninfected mice (*p* ≤ 0.0001 and *p* ≤ 0.01 respectively)
				Histological and immunofluorescence assessment		
Anti-inflammatory agent	Napimoga et al., 2012 [28]	15d-PGJ2-loaded poly (D,L-lactide-coglycolide) (PLGA) nanocapsules (NC)	Experimental periodontitis in mice induced by oral delivering of *A. actinomycetemcomitans JP2* with a micropipette 3 groups (*n* = 5/group): uninfected, infected and saline, infected and 15d-PGJ2-NC	Dissecting microscope	Test the efficacy of PLGA nanocapsules loaded with 15d-PGJ2 in bone loss and immunoinflammatory responses	**After 15 days**: The treatment by 15d-PGJ2 NC at 10 µg/kg showed significantly lower ABL (0.07 mm) compared to infected and saline group (0.12 mm) (*p* < 0.05) The gingival tissues of 15d-PGJ2 NC treated group showed significantly: lower submandibular lymph nodes infiltration by lymphocytes (CD4, CD8) (*p* < 0.05), lower RANK-L and mRNA (IL-17, IL-6, FOXP3, CCL-22, IL-10, TGF-ß) expression and MPO activity compared to PBS group (*p* < 0.05) and higher CD55 expression and 15d-PGJ2 amount compared to PBS group (*p* < 0.05) No significant differences between 15d-PGJ2-NC and infected and saline groups were shown, in terms of IgG anti-*A. actinomycetemcomitans* production, OPG, IL-15 mRNA expression
				Elisa immunoassays		
				Histomorphometrical assessments		
				Flow cytometry analysis		
				Spectrophotometry measurement		
				Western-blotting		
				RT-PCR		
	Srivastava et al., 2014 [29]	Nanoemulgel (NEG) of ketoprofen (KP) containing eugenol	Experimental periodontitis (EPD) in rats induced by ligatures 4 groups (*n* = 6/group): control (no EPD), ligature only, NEG and ligature, KP loaded NEG and ligatures (NEG + KP)	Evaluation of clinical periodontal parameters: gingival index (GI), tooth mobility (TM)	Assess the efficacy of KP loaded NEG containing eugenol	**After 11 days**: The NEG+KP treatment showed significantly lower ABL (0.14 mm) compared to the ligature only group (0.68 mm) (*p* < 0,05) The NEG+KP treatment also showed significantly lower GI, TM (*p* < 0,05) and cytokines expression (IL-1β, TNF-α) Histopathology of the periodontium showed a reduced inflammatory cell infiltration, alveolar bone and cementum resorption for the NEG + KP group, in comparison with ligature only group AFM showed a reduction of roughness of alveolar bone surface in the NEG+KP group, in comparison with ligature only group
				Histopathological assessment		
				Elisa immunoassays		
				Dissecting microscope		
				Atomic force microscopy		
	Saita et al., 2016 [30]	Redox injectable gel (RIG) from nitroxide radicals containing polyion complex (PIC) flower micelles	Experimental periodontitis in rats induced by oral delivering of *P. gingivalis* 4 groups (*n* = 14/group): control (sham, without infection), *P. gingivalis*, noRIG+*P. gingivalis (radical without nitroxide)*, RIG+*P. gingivalis*	Gingival blood flow measure	Evaluate the ROS-scavenging antioxidant-related anti-inflammatory and anti-*P. gingivalis* induced bone loss effects of the RIG	**After 21 days**: The amount of ABL was significantly lower in the RIG+*P. gingivalis* group (0.341 ± 0.035 mm) and control group (0.310 ± 0.026 mm) than in the *P. gingivalis* (0.479 ± 0.038 mm) and nRIG+*P. gingivalis* groups (0.470 ± 0.036 mm) (*p* < 0.01, *n* = 6) The gingival blood flow rate was significantly decreased in *P. gingivalis* and noRIG+*P. gingivalis* rats compared to the sham group (*p* < 0.05), while the RIG+*P. gingivalis* rats retained the same level of gingival blood flow as that of the control (sham) rats (*n* = 3) noRIG did not effectively suppress oxidative stress induced by *P. gingivalis*, while the MDA levels in the RIG group were consistent with significant suppression of oxidative stress at day 21 (*p* < 0.05, *n* = 4) No significant changes were observed in the levels of IL-1β and IL-6 The number of osteoclasts (TRAP) in the *P. gingivalis* and noRIG+*P. gingivalis* groups was significantly higher than that observed in the control and RIG+*P. gingivalis* groups (*p* < 0.001, *n* = 6)
				Histological assessment		
				Elisa immunoassays		
				µCT (computed tomography) analysis		
	Khajuria et al., 2017 [31]	Chitosan-based risedronate/zinc-hydroxyapatite nanoparticles intrapocket dental film (CRZHDF)	Experimental periodontitis in rats induced by gingival injections of *P. gingivalis*-LPS 5 groups (*n* = 12/group): healthy, untreated periodontitis, periodontitis + CRZHDF-0.1, periodontitis + CRZHDF-0.2, periodontitis + chitosan film	Elisa immunoassays	Develop a CRZHDF for applications in the treatment of ABL	CRZHDF reversed alveolar bone resorption (ABL: CRZHDF-0.1: 19.23 ± 0.61 mm; CRZHDF-0.2: 21.61 ± 0.38 mm) when compared to the untreated periodontitis group (ABL: 38.10 ± 0.88mm) (*p* < 0.001) CRZHDF also resulted in significant improvements in the mesial and distal periodontal bone support proportions (%) and bone mineral density b-ALP activity and TRACP 5b were lower in CRZHDF groups compared to the untreated periodontitis group (*p* < 0.0001) The expression of OCN was higher in CRZHDF groups compared to the untreated periodontitis group (*p* < 0.0001)
				µCT analysis		
				Histological assessment		
	Pereira et al., 2018 [32]	Metformin hydrochloride-loaded nanoparticles poly(D,L-lactide-co-glycolide) (MET-loaded PLGA)	Experimental periodontitis (PD) in diabetes (DM) rats induced by ligatures 9 groups (*n* = 20/group): sham (without DM, without PD), PD without DM, DM without PD, positive control with PD and DM, PLGA control with PD and DM (*n* = 6), Met50, Met100, MET-loaded PLGA 100, MET-loaded PLGA 10	Elisa immunoassays	Evaluate the effect of MET-loaded PLGA	**After 11 days**: Bone loss was reduced when comparing positive control (0.97 ± 0.35 mm) to PLGA 10 mg/kg MET treatment (0.65 ± 0.14 mm) (non-significant) Treatment with MET-loaded PLGA 10 mg/kg showed: low inflammatory cells; weak staining by RANKL, cathepsin K, OPG, and OCN; reduced levels of IL-1β and TNF-α; increased AMPK expression gene; and decreased of NF-KB p65, HMGB1 and TAK-1 (*p* < 0.05)
				qRT-PCR		
				Histopathological and immunohistochemical assessment		
				µCT analysis		
	Zambrano et al., 2018 [33]	Polylactic and polyglycolic acids nanoparticles (NP) loaded with curcumin	Experimental periodontitis in rat induced by gingival injections of *E.Coli*-LPS 4 groups (*n* = 4/group): PBS-empty NP, LPS-empty NP, PBS-curcumin NP, LPS-curcumin NP	µCT analysis	Assess the biological effect of the local administration of curcumin in a nanoparticle vehicle	**After 28 days**: Curcumin NP resulted in an inhibition of inflammatory bone resorption: PBS-curcumin NP and LPS-curcumin NP have a bone volume/total volume (BV/TV) of 64 and 65% respectively (non-significant); PBS-empty NPand LPS-empty NP have a BV/TV of: 65% and 47% respectively (*p* < 0.05)) Curcumin NP also resulted in a decrease of both osteoclast counts (*p* < 0.001) and inflammatory infiltrate (*p* < 0.05); as well as in a marked attenuation of p38 MAPK and NF-KB activation
				Histomorphometric analysis		
				Western blot		
	Ni et al., 2019 [34]	Gold nanoparticles (AuNP) coated with L-cysteine	Experimental periodontitis in rat induced by ligatures 3 groups (*n* = 6/group): control (no ligature), ligatures, ligatures+AuNP (Lig-AuNP)	µCT analysis	Evaluate the potential application of AuNP	**After 14 days**: The injection of AuNP could significantly alleviate the ABL surrounding the maxillary second molars caused by ligatures: Lig 0.69 mm vs. Lig-AuNP 0.38 mm and control 0.34mm (*p* < 0.001) The elastic and collagenous fibers were denser and more well-organized in the groups with Lig-AuNP The number of osteoclasts was decreased by AuNP when the ligation existed (*p* < 0.001) The AuNP could inhibit this inflammatory response and downregulate the level of iNOS (*p* < 0.01)
				Histological and immunohistochemical assessment		
	Valerio et al., 2019 [35]	Polyethylene glycol (PEG)-polylactide (PLA) (PEG-PLA) nanoparticles loaded with Auranofin (ARN)	Experimental periodontitis in rats induced by injections of *A. actinomycetem* *comitans*-LPS 4 groups (*n* = 10/group): PBS alone, NP only (no ARN), NP-ARN high (10 µM), NP-ARN low (1 µM)	µCT analysis	Determine if nanoparticles loaded with a pharmacological agent that induces mitogen-activated protein kinase phosphatase has potential clinical utility for management of ABL	**After 14 days**: NP-ARN low was significantly effective at inhibition of LPS-induced bone loss compare to PBS (BV/TV 24% vs. 31% respectively)
Anti-inflammatory and antibacterial agent	Lin et al., 2017 [36]	Polylactide-glycolic acid co-polymer and chitosan (PLGA/chitosan) with metronidazole or N-phenacyl-thiazolium bromide (PTB)	Experimental periodontitis in rat induced by ligatures 4 groups (*n* = 4/group): periodontitis alone (PR), periodontitis with nanospheres alone, nanospheres encapsulating metronidazole (MT), nanospheres encapsulating PTB (PB)	µCT analysis	Develop pH-responsive PLGA/chitosan nanosphere as an inflammation-responsive vehicle Evaluate the potential of the nanosphere encapsulating metronidazole, an antibiotic, and N-phenacylthiazolium bromide (PTB), a host modulator	**After 21 days**: Progression of periodontal bone loss (PPBL) was significantly reduced in groups MT (−0.1 mm) and PB on day 4 (−0.17 mm) compared with group PR (0.03 mm) (*p* < 0.05). On day 21, PPBL was significantly lower in group PB (−0.04 mm) compared with group PR (0.13 mm) and group MT (0.07 mm) (*p* < 0.05) On day 21, inflammation was significantly reduced in groups MT and PB relative to groups PR and periodontitis with nanospheres alone (*p* < 0.05), and collagen deposition was greater relative to group PR (*p* < 0.05)
				Histological assessment		

Abbreviations: *P. gingivalis*: *Porphyromonas gingivalis*, *P. intermedia*: *Prevotella intermedia*, *S. gordonii*: *Streptococcus gordonii*, *A. actinomycetemcomitans*: *Actinobacillus actinomycetemcomitans*, *E. coli*: *Escherichia Coli*, LPS: lipopolysaccharides, ABL: alveolar bone loss, AMPK: AMP-activated protein kinase, b-ALP: bone alkaline phosphatase, BV/TV: bone volume/total volume, CCL-22: C–C motif chemokine ligand 22, CD: cluster of differentiation, FOXP3: forkhead box P3, HMGB1: high-mobility group box 1, Ig: immunoglobulin, IL: interleukin, iNOS: inducible nitric oxide synthase, MAPK: mitogen-activated protein kinase, MDA: malondialdehyde, MPO: myeloperoxidase, n: number of animals, NFKB: nuclear factor kappa B, OCN: osteocalcin, OPG: osteroprotegerin, RANKL: receptor activator of nuclear factor kappa-B ligand, TAK-1: TGF-b–activated kinase 1, TGF-β: transforming growth factor β, TNF-α: tumor necrosis factor α, TRACP 5b: tartrate-resistant acid phosphatase 5b, TRAP: tartrate-resistant acid phosphatase.

**Table 2 nanomaterials-10-01414-t002:** The reported nanoparticles: characteristics, dose and cytotoxicity.

Treatment Type	Authors, Year	Nanoparticle	Coating/Loading	Shape	Size (Mean ± SD)	Dose	Cytotoxicity	Degradability
Antibacterial agent	Botelho et al., 2010 [26]	Nanostructured doxycycline gel	/	Spherical	Nanometer scale	1 g	The nanoparticle is known to be biocompatible*	Known to be biodegradable *
	Mahmoud et al., 2019 [27]	Poly(lactic-co-glycolic acid) (PLGA)	BAR peptide	Spherical	87.9 ± 29.4nm (unhydrated) 333.8 ± 17.8 nm (hydrated)	0.7 µM	BNPs were non-toxic within the evaluated concentration range of 1.3–3.4 μM. Telomerase immortalized gingival keratinocytes treated with BNPs or free BAR demonstrated > 90% viability and no significant lysis or apoptosis relative to untreated cells In addition, neither BNPs nor free BAR exhibited haemolytic activity	Biodegradable
Anti-inflammatory agent	Napimoga et al., 2012 [28]	PLGA	15-Deoxy-D12,14-PG J2 (15d-PGJ2)	Spherical	Nanometer scale	/	The nanoparticle is known to be biocompatible *	Biodegradable
	Srivastava et al., 2014 [29]	Eugenol	Ketoprofen	Spherical	37.230 ± 0.210 nm	/	The nanoparticle is known to be biocompatible *	Known to be biodegradable *
	Saita et al., 2016 [30]	Polyon complex composed of (PMNT-PEG-PMNT) triblock copolymer and anionic poly(acrylic acid)	/	Flower-like micelle	79 nm	/	The nanoparticle is known to be biocompatible *	Known to be biodegradable *
	Khajuria et al., 2017 [31]	Zinc-hydroxyapatite (chitosan-based)	/	/	Nanometer scale	0.2% w/v	The nanoparticle is known to be biocompatible *	Known to be biodegradable *
	Pereira et al., 2018 [32]	PLGA	Metformine hydrochloride	Spherical	457.1 ± 48.9 nm	/	The nanoparticle is known to be biocompatible *	Biodegradable
	Zambrano et al., 2018 [33]	PLGA	Curcumin	/	Nanometer scale	/	The nanoparticle is known to be biocompatible *	Known to be biodegradable*
	Ni et al., 2019 [34]	Gold (Au)	L-cysteine	/	45 nm	0.25 µM	AuNP did not show any significant cytotoxicity on mouse macrophage cell line (cell viability, membrane integrity, ROS production assays) AuNPs did not affect cell viability of murine bone marrow-derived macrophage	Non-degradable
	Valerio et al., 2019 [35]	PEG-PLA	Auranofin (ARN)	Spherical	Nanometer scale	1 or 10 µM	- ARN-NP did not significantly affect cell viability of murine macrophage, in contrast to higher doses of free ARN	Biodegradable
Anti-inflammatory and antibacterial agent	Lin et al., 2017 [36]	PLGA and chitosan	Metronidazole or *N*-phenacyl-thiazolium bromide	Spherical	499 ± 21.24 nm	/	The nanoparticle is known to be biocompatible *	Biodegradable

* According to the literature.

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
