# Peer review of "Emerging Nanotechnology in Non-Surgical Periodontal Therapy in Animal Models: A Systematic Review"

_nanomaterials, 2020, doi:10.3390/nano10071414_

Round 1

Reviewer 1 Report

The paper is overall well designed and conducted.

In the introduction (r. 82) please also cite a recent review published on Nanomaterials (Carrouel Fet al. Nanoparticles as Anti-Microbial, Anti-Inflammatory, and Remineralizing Agents in Oral Care Cosmetics: A Review of the Current Situation. Nanomaterials (Basel). 2020 Jan 13;10(1)) which refers to updated review of NPs as Anti-Microbial, Anti-Inflammatory agents.

In the conclusions, the term “global” (r. 413) seems to be overstated, since it is referred to existing encouraging literature, and is clearly defined as “lacking of strong evidence” by the authors themselves at r. 419. Please redefine the sentence to relativize the statement. Moreover it should be underlined that, along with the clinical efficacy, further evidence is needed due to the safety concerns (as correctly explained in the toxicity paragraph at r.362-368).

Author Response

Concern 1: The paper is overall well designed and conducted.

>> Our response: Thank you for this general positive comment. We have carefully revised our manuscript to address your concerns.

Concern 2: In the introduction (r. 82) please also cite a recent review published on Nanomaterials (Carrouel Fet al. Nanoparticles as Anti-Microbial, Anti-Inflammatory, and Remineralizing Agents in Oral Care Cosmetics: A Review of the Current Situation. Nanomaterials (Basel). 2020 Jan 13;10(1)) which refers to updated review of NPs as Anti-Microbial, Anti-Inflammatory agents.

>> Our response: We appreciate this suggestion that reinforce our introduction. We have added the reference, as follows: “Various nanoparticulate systems have been investigated for the treatment of periodontitis. [Brun et al, 2019; Carrouel et al., 2020]” (lines 82 and 83).

Concern 3: In the conclusions, the term “global” (r. 413) seems to be overstated, since it is referred to existing encouraging literature, and is clearly defined as “lacking of strong evidence” by the authors themselves at r. 419. Please redefine the sentence to relativize the statement. Moreover it should be underlined that, along with the clinical efficacy, further evidence is needed due to the safety concerns (as correctly explained in the toxicity paragraph at r.362-368).

>> Our response: We agree the comment and have modified our conclusion as follows: “The results herein show that nanoparticles used in periodontal indications may have a positive effect on alveolar bone loss in preclinical studies. High nanoparticles sustainability with an extended release would be of crucial interest. Agents modulating inflammation seems to be more relevant than antibiotics, in terms of efficiency and risk of antibiotic resistance. Moreover, poly(lactic-co-glycolic acid) or drugs used as their own carrier appear to be the most interesting nanoparticles in terms of biocompatibility.

Finally, despite lack of strong evidence on their clinical efficacy, there are encouraging preclinical data of using nanoparticles as a contribution to the treatment of periodontitis. Further evidence is however needed due to the safety concerns.” (lines 415 to 423).

Reviewer 2 Report

Dear authors,

Thank you for submitting the manuscript entitled “Emerging nanotechnology in non-surgical periodontal therapy in animal models: a systematic review”. The topic is of interest, but several issues have been highlighted.

Research question - From the title, the aim of your work appears to be the role of nanotechnology in non-surgical periodontal therapy. Do you mean as an “adjunct to”? Or a as treatment alone? This is unclear.

Definition of the intervention - Also, I could not find in the manuscript details of what non-surgical therapy actually was (how was conducted) in the included studies – this should be specified.

PICOS

Comparator/control: untreated animals

To answer your research question, you should have also included a comparator consisting of non-surgical periodontal therapy alone?

Outcome: improvement of alveolar bone level

The outcome is too vague – what do you mean with “improvement”? Bone density? Change in bone levels?

Literature search

Please report timeframe of search conducted.

Outcomes measures

Decrease of alveolar bone loss (ABL) – this is different from the PICO.

This should be better defined – how is it measured?

What is the follow-up of included studies? This is not reported.

There is no mention of summary measures used and how synthesis of data was performed. Although meta-analysis could not be performed, summary measures and unit of analysis should be defined.

Discussion

“The overall results of the present systematic review showed that nanoparticles have a positive effect in non-surgical periodontal therapy” – this can not be really concluded. In fact you highlight later the main limitations of included studies:

“Besides, none of the studies compared the use of nanoparticles to the gold standard of periodontal treatment, i.e. mechanical debridement, alone or associated with chemotherapeutic agents, which would attest the benefit of nanoparticles.’

Conclusions

“The results herein show that nanoparticles used in periodontal indications have a global positive effect on the outcomes of non-surgical periodontal treatment in preclinical studies. High nanoparticles sustainability with an extended release would be of crucial interest.”

Again, you might need to review your conclusions on the basis of what highlighted above.

Regards

Author Response

Dear authors,

Concern 1: Thank you for submitting the manuscript entitled “Emerging nanotechnology in non-surgical periodontal therapy in animal models: a systematic review”. The topic is of interest, but several issues have been highlighted.

>> Our response: Thank you for this comment, which has led us to improve our manuscript.

Concern 2: Research question - From the title, the aim of your work appears to be the role of nanotechnology in non-surgical periodontal therapy. Do you mean as an “adjunct to”? Or a as treatment alone? This is unclear.

>> Our response: The aim of our study was to investigate the role of nanotechnology as a treatment alone. We have now specified that:

- in the abstract, as follows: “The aim of this systematic review was to investigate the effect of nanoparticles as a treatment alone in non-surgical periodontal therapy in animal models.” (lines 23 to 25)

- and in the introduction, as follows: “Hence, the aim of this study was to systematically review the effect of nanotechnology-based drug systems as a treatment alone in the non-surgical treatment of periodontitis in preclinical animal models.” (lines 86 to 88).

Concern 3: Definition of the intervention - Also, I could not find in the manuscript details of what non-surgical therapy actually was (how was conducted) in the included studies – this should be specified.

>> Our response: We have now specified it in the PICO question: “Intervention/exposure: use of nanoparticles systems as non-surgical periodontal therapy (i.e. injected in the periodontal pocket without involving incisions and flap elevation)” (lines 101 and 102).

Concern 4: PICOS

Comparator/control: untreated animals

To answer your research question, you should have also included a comparator consisting of non-surgical periodontal therapy alone?

>> Our response: Thanks to the previous comment, the aim of our work is now more accurate. Indeed, the aim of this systematic review was to investigate the effect of nanoparticles as a treatment alone in non-surgical periodontal therapy in animal models (lines 86 to 88). Thus, to assess if nanoparticles had an impact on the periodontitis, we included studies evaluating the effect of nanoparticles on treated animals compared to untreated animals.

However, as you pointed out it would be relevant to compare the efficacy of nanoparticles as a treatment alone to the gold standard non-surgical therapy (mechanical debridement). We precisely raised that in the discussion with the following sentence: “Besides, none of the studies compared the use of nanoparticles to the gold standard of periodontal treatment, i.e. mechanical debridement, alone or associated with chemotherapeutic agents, which would attest the benefit of nanoparticles” (lines 286 to 290).

Concern 5: Outcome: improvement of alveolar bone level

The outcome is too vague – what do you mean with “improvement”? Bone density? Change in bone levels?

>> Our response: We agree this comment and we have specified the outcome as: “decrease of alveolar bone loss” (line 104).

Concern 6: Literature search

Please report timeframe of search conducted.

>> Our response: We have added in the materials and methods section the timeframe of search conduced, as follows: “A comprehensive literature search was performed on the electronic databases Medline via PubMed, Web of Science, the Cochrane Library and ScienceDirect, up to June 2019” (lines 107 and 108).

Concern 7: Outcomes measures

Decrease of alveolar bone loss (ABL) – this is different from the PICO.

>> Our response: We agree the comment. As noted in the concern 5, we have corrected the outcome of the PICO question as: “decrease of alveolar bone loss” (line 104).

Concern 8: This should be better defined – how is it measured?

>> Our response: We have modified the outcomes as follows: “The primary outcome measure was a decrease of alveolar bone loss (ABL) evaluated macroscopically, radiographically (µCT computed tomography) or microscopically (dissecting microscope).” (lines 131 to 133).

Concern 9: What is the follow-up of included studies? This is not reported.

>> Our response: The follow-up of included studies has now been added in the results section: “The follow-up of the studies was up to 28 days.” (lines 173 and 174). It was only specified in the “Main results” column of Table 1. We have now highlighted it in bold.

Concern 10: There is no mention of summary measures used and how synthesis of data was performed. Although meta-analysis could not be performed, summary measures and unit of analysis should be defined.

>> Our response: The units of ABL were only specified in the “Main results” column of Table 1. It has now been added in the Results section: “One study reported lower GI and TM in rats. All the studies reported a decrease in ABL (expressed in mm or in %) and 3 studies also showed a decrease in osteoclast count.” (lines 232 and 233).

Summary measures were compiled in the “Main results” column of Table 1. The synthesis was expressed in the “Main results” (3.5) paragraph of the Results and discussed in the “Methodological heterogeneity and limitations” paragraph of the Discussion.

The data were compared between the studies. ABL was analysed when it was expressed in mm. We have now specified that in the beginning of the Discussion as follows: “The data of the studies were compared as possible. ABL was analysed when it was expressed in mm” (lines 278 and 279).

The data were qualitatively discussed, as follows: “We can note that the difference in ABL between treatment and control group, is higher for BAR peptide and doxycycline than metronidazole” (lines 316 and 317) and “ABL seems to be generally lower when using anti-inflammatory” (lines 342 and 343).  

Concern 11: Discussion

“The overall results of the present systematic review showed that nanoparticles have a positive effect in non-surgical periodontal therapy” – this can not be really concluded. In fact you highlight later the main limitations of included studies:

“Besides, none of the studies compared the use of nanoparticles to the gold standard of periodontal treatment, i.e. mechanical debridement, alone or associated with chemotherapeutic agents, which would attest the benefit of nanoparticles.’

>> Our response:  We agree the comment, the discussion has been changed to be more accurate: “The overall results of the present systematic review showed that nanoparticles may have a positive effect to prevent alveolar bone loss in periodontitis animals.” (lines 279 and 280).

Concern 12: Conclusions

“The results herein show that nanoparticles used in periodontal indications have a global positive effect on the outcomes of non-surgical periodontal treatment in preclinical studies. High nanoparticles sustainability with an extended release would be of crucial interest.”

Again, you might need to review your conclusions on the basis of what highlighted above.

Regards

>> Our response: We agree the comment, we have modified the conclusion as follows: “The results herein show that nanoparticles used in periodontitis may have a positive effect on the alveolar bone loss in preclinical studies.” (lines 415 and 416).